# Graph Based Neural Networks for Interrogating Multiplexed Tissue Samples

Anonymous[1]

[1] Anonymous Institution
[2] Anonymous Institution

**Abstract.** Multiplexed immunofluorescence provides an unprecedented opportunity for studying specific cell-to-cell and cell-microenvironment interactions. Noise, imaging artifacts, and the variation in protein expression make this a particularly challenging problem. We employ graph neural networks to combine features obtained from tissue morphology with measurements of protein expression to identify communities of cells related to tumour stage. Our framework presents a new approach to analysing and processing these complex multi-dimensional datasets.

**Keywords:** Multiplexed Immunofluorescence · Graph Neural Networks · Digital Pathology · Colorectal Cancer.

## 1 Introduction

Novel tissue multiplexing imaging platforms [20, 1] allow the analysis of a broad range of cell types in the tissue architecture context. These approaches open up new opportunities for improving our understanding of disease, monitoring therapeutic response, and the development of high-dimensional clinical tests. Here, we are interested in predicting tumour progression which is strongly dependent on the tumour microenvironment (TME) and the complex interaction between the developing tumour and the immune system. While current cancer classification highly relies on the extent of the primary tumour (T), lymph node involvement (N) and metastatic presence (M), visualising multiple protein targets in the same tissue allows us to interrogate the involvement of and the role of adaptive immune cell infiltration in colorectal cancer (CRC) prognosis.

The analysis of multiplexing data requires the combination of spatial information that captures the changes in tissue architecture with measurements of protein expression. When compared to standard digital pathology, multiplexing datasets are typically much smaller. While techniques developed for cytometry have been applied to these datasets, the inherently complex statistics pose challenges that require a more principled approach. Building on recent success of applying graph neural networks (GNNs) to histopathology, we introduce a novel framework for analysing multiplexed immunofluorescence images using GNNs that involves selecting regions of interest, combining cell and tile level features, and augmenting the size of the training data. To further investigate immune-cell interactions, we introduce a set of network metrics.

In summary, we propose a GNN model to predict tumour stage in an explainable setting that allows us to discover which parts of the tissue contribute to the prediction. To the best of our knowledge, this is the first attempt at predicting tumour stage from multiplexed images using graph-based neural networks.

## 2   Methods

The two-layer graph described in Section 2.1 is constructed to abstract the key parameters of the underlying tissue. It captures the locations of cells, certain morphological measurements, protein expression, and selected immune-interaction features. Rather than performing a global analysis of the graph, we perform a local analysis in selected regions of interest (RoIs). A GNN is used to predict tumour stage for a given RoI-level-graph. Finally, post-hoc explainability methods, presented in Section 2.4 are utilised to visualise the relationship between immune interaction profiles and prediction. Figure 1 provides a summary of the overall approach.

### 2.1   Graph representation

We define an undirected graph $G = (V, E)$, with vertices $V$ and edges $E$. The graph topology is represented by an adjacency matrix $A \in \mathbb{R}^{N \times N}$, and node features are represented by the matrix $X \in \mathbb{R}^{N \times D}$, with $N = |V|$ and feature dimension $D$. Similar to Pati *et al.* [14] we employ a two-layer graph representation, with (1) cell-graphs [4] constructed on small randomly sampled tiles, in order to quantify local patterns of immune interaction, and (2) a tile-level graph able to aggregate information from the multiple tissue regions. We construct $A$ based on a distance threshold as follows:

$$A_{ij} = \begin{cases} 1 \text{ if } d(i,j) < k \\ 0 \text{ otherwise,} \end{cases} \tag{1}$$

where the threshold $k$ which determines whether two nodes are connected or not is selected so as to ensure a small average node degree (the number of edges connected to each node) as well as a small number of disconnected nodes to reduce graph complexity and facilitate metric computation.

### 2.2   Network metric computation

Network metrics are used to investigate the distribution of each cell population of interest. These metrics can be applied at the cell-graph level, to acquire information about the general distribution of the nodes, or to subgraphs, where only nodes of a specific target of interest (e.g. helper T-cells) and their respective edges are selected, keeping their original node positions intact. The metrics include the average clustering and square clustering coefficients, the assortativity, radius, density, transitivity, and the closeness of each cell type population, as defined by Kerem *et al.* [18]. The ratios between each pair of immune cell densities,

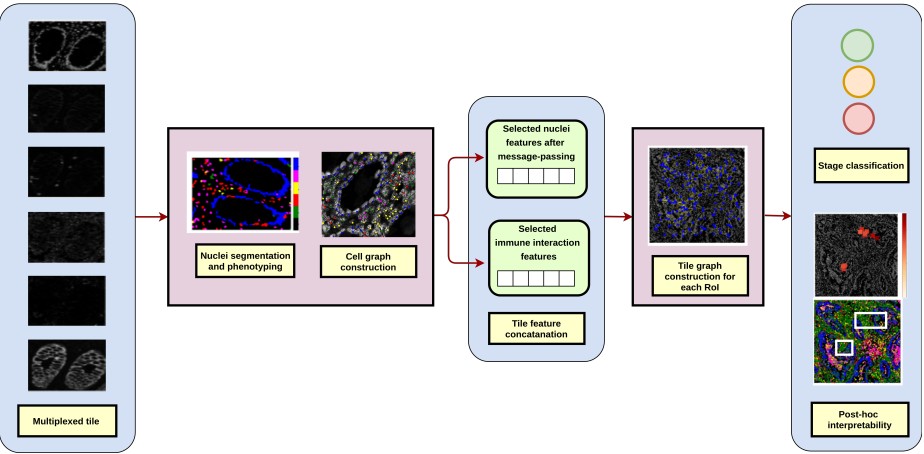

**Fig. 1.** Overview of the proposed method.

a known prognostic factor for cancer progression [19, 7, 5] are also computed. To measure the degree of mixing between tumour and immune cells, we additionally compute the ratio of immune-tumour to immune-immune interactions [9].

### 2.3   Graph Neural Networks

We employ Graph Neural Networks (GNNs) to obtain a graph representation $H \in \mathbb{R}^{N \times P}$ from our initial embedding $H^0 = X \in \mathbb{R}^{N \times D}$, where $P$ is the number of output features. Using the notation from [10] and [23], we first perform a number of message passing steps to obtain the node embeddings $h_v$ for each node $v \in G$, which we then combine into a global graph embedding $H_G$ by means of a readout layer. The message passing consists of an aggregation and combination of the neighbouring nodes features. For the $k^{th}$ GNN layer:

$$a_v^{(k)} = \text{AGG}^{(k)} \left( \left\{ h_u^{(k-1)} : u \in \mathcal{N}(v) \right\} \right), \ h_v^{(k)} = \text{COMBINE}^{(k)} \left( h_v^{(k-1)}, a_v^{(k)} \right)$$
$$(2)$$

$$h_G = \text{READOUT} \left( \left\{ h_v^{(k)} \mid v \in G \right\} \right), \tag{3}$$

where $\mathcal{N}(v)$ denotes the set of neighbours of $v$. For the message passing step, we will use the graph convolution operator defined in [13], which preserves central node information and improves the characterisation of real-world networks [3]:

$$h_v^{(k)} = W_1^{(k)} h_v^{(k-1)} + W_2^{(k)} \sum_{u \in \mathcal{N}(v)} h_u^{(k-1)}. \tag{4}$$

Following the nuclei feature message passing in the cell-graphs $G_t$, we employ a mean readout layer. Combined node features and selected network metrics $m_t$

listed in the previous sub-section are then concatenated for each tile $t$:

$$h_{\mathrm{t}}^{(0)} = \mathrm{CONCAT}\left(m_{\mathrm{t}}, h_{G_t}\right). \tag{5}$$

For the RoI-level tile-graph, we apply Eq. 4 again for each RoI, with nodes corresponding to the individual tiles. We compare the results using mean, sum, and max global pooling.

### 2.4   Post-hoc explainability

We employ Integrated Gradients (IG) [21, 8] to understand the significance of each tile node in predicting tumour stage. We do so by computing the IG attribution of each edge and aggregating the attributions of the edges connecting each node. The IG edge attribution is computed by comparing each edge mask with a baseline of edge weights set to zero. Since we use unweighted graphs, the initial edge weights are all one. The IG for each edge $e_i$ is computed as follows:

$$\mathrm{IG}_{e_i} = \int_{\alpha=0}^{1} \frac{\partial F(x_{\alpha)}}{\partial w_{e_i}} d\alpha, \tag{6}$$

where $x_\alpha$ corresponds to the original input graph but with all edge weights set to $\alpha$, $w_{e_i}$ denotes the current edge weight, and $F(x)$ is the output of the model for an input $x$. The integral can be approximated using Gauss-Legendre quadrature.

In order to identify the key features impacting the prediction, we further run the GNN Explainer model [24], which maximises the mutual information MI between the prediction of the trained model and that of the simplified explainer model given by a subgraph $G_S$ and a subset of features $T$: $\max_{G_S, T} \mathrm{MI}(Y, (G_S, T))$.

## 3   Experiments

**Dataset and marker panel.** Paraffin-embedded tissue samples of 41 rectal primary tumours were used to investigate the risk of disease progression and recurrence. All samples were processed on the Perkin-Elmer Vectra platform using an immune panel of 6 fluorescent markers. DAPI is used for nuclei segmentation. Cytokeratin (Opal 650) is used to delineate epithelial cells. A further four markers are included to visualise immune cells: CD4 (Opal 520) for helper T-cells, CD8 (Opal 570) for cytotoxic T-cells, CD20 (Opal 540) for B-cells, and Foxp3 (Opal 620) for regulatory T-cells. The system also provides a seventh channel corresponding to the imaging system's autofluorescence isolation capacity which improves signal-to-noise ratio [22].

Specialist GI pathologists reported tumour stage on matching H&E slides: 28 of these samples were assigned a pT1 tumour stage, while 16 samples were considered to be more advanced (13 pT2, 3 pT3). Specific regions of interest such as those shown in Fig. 2 were provided by a pathologist for the tumour centre, invasive tumour front, background mucosa, and peritumoural stroma, guided by the matched H&E image. Annotation areas correspond to the standard 1mm diameter disk size used for biopsies and tissue microarrays (TMAs), allowing for a future integrative analysis with TMA cohorts.

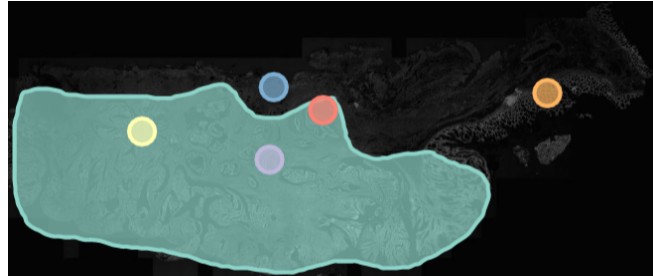

**Fig. 2.** Example of a WSI with annotations for the tumour centre (purple and yellow), invasive front (red), background mucosa (orange), and peritumoural stroma (blue).

**Nuclei segmentation and cell phenotyping.** Before segmentation, a background and shading correction [15] was performed to improve the stitching of individual image tiles. Moreover, contrast limited adaptive histogram equalisation (CLAHE) was used to improve contrast in the DAPI channel. The segmentation network employed to identify cell nuclei consists of a 3-class (cell inside, cell boundary, background) modified U-Net architecture comprised of the original U-Net [16] decoder and a ResNet [6] encoder, where downblocks are replaced by residual blocks. The model was pre-trained on the fluorescence samples from the publicly available BBBC038v1 dataset (Broad Institute Bioimage Benchmark Collection) [2, 12]. On average, the network identified a total of 1.3M cells per sample. Segmentation masks were projected onto the remaining channels to measure the average nuclei protein expression. Cells were then assigned to the cell type corresponding to the validated marker with the highest percentile allocation by applying quantile normalisation.

**Graph construction** RoIs of the size of 2048x2048 pixels corresponding to the bounding box of the disk annotations are selected to investigate immune-cell interactions across samples and regions. From each RoI, 200 potentially overlapping 256x256 tiles are randomly chosen to construct cell-graphs. A minimum threshold of 50 cells per tile is set to avoid sampling from predominantly background tiles. The total number of tiles available for each RoI is shown in Table 1, together with the median number of cells per tile.

Cell-graph networks for each tile are constructed using NetworkX [18] with nodes centered at the centroid of each nucleus. The adjacency matrix is built based on the distance between nuclei. We choose a threshold of 30 pixels, which leads to a node degree of 8.1 and 76% of the nodes in the largest connected component on average. For each node, we record the average marker expression for the five markers of interest, the area occupied by the cell, and the cell solidity. These seven features are inserted as node features. We subsequently perform three message-passing steps to update the node features by encompassing information from nearby cells, which are aggregated using mean pooling and transformed into a vector of length 16. Additionally, for each tile, we compute

the set of 68 hand-crafted immune-interaction features enumerated in Section 2.2. Nuclei and cell-interaction features are then concatenated into a vector of length 84 per tile. The second set of graphs are constructed at the RoI level, with nodes representing the 200 sampled tiles positioned at their tile centre coordinates. Each tile incorporates the selected 84 attributes as node features. These RoI-level graphs are then fed into the model for pT stage prediction.

**Table 1.** Number of RoI images, tiles sampled, and number of cells (median and interquantile range) available from each region and tumour stage.

| Region / Stage | # Tiles | # RoIs | Median # Cells per tile | Median # Cells per RoI |
|---|---|---|---|---|
| Centre | 16,400 | 82 | 123; IQR: (92, 158) | 7232 (6387, 9377) |
| Front | 8,200 | 41 | 108; IQR: (80, 142) | 6185 (4717, 8065) |
| Mucosa | 6,200 | 31 | 109; IQR: (82, 140) | 5433 (4156, 7032) |
| Stroma | 7,800 | 39 | 76;  IQR: (61, 103) | 3293 (2652 , 4627) |
| pT1 | 23,600 | 118 | 112; IQR: (79,152) | 6619 (4239, 8325) |
| pT2 | 12,200 | 61 | 106; IQR: (78, 137) | 6394 (4371, 7425) |
| pT3 | 2,800 | 14 | 90;  IQR: (68, 125) | 5240 (3889, 6561) |
| All | - | - | 108; IQR: (77, 145) | 6385 (4229, 7923) |
| Total | 38,600 | 193 | 1,207,505 | 1,207,505 |

**Data augmentation.** For each RoI in the training set, a 150x pre-batching augmentation strategy is employed to reduce over-fitting. The augmented set is obtained by constructing the networks using a subsample of 80% of the nodes at each step (160 tiles) and by varying the threshold that needs to be surpassed for an edge to be included between two neighbours by sampling a value in the range {150, 175, 200, 225, 250}, resulting in a variety of tighter and sparser graphs. The node subsampling and edge modifications ensure that networks in the training set are sufficiently different to avoid over-fitting. For the test set, only a single network is constructed per RoI using the default distance threshold of 200 pixels for adjacency construction and the full set of tile nodes in the RoI (200 tiles).

**Implementation.** The model consists of three GraphConv [13, 3] layers with ReLu activation and global pooling aggregation. Experiments are conducted in PyTorch 1.7.0 using PyTorch Geometric [3] on a local machine with 16GB RAM on CPU-only mode. Data is split into training and testing at the patient level. Due to the limited sample size, a pseudo-validation set is constructed by randomly sampling (with pT stratification) 10% of the pre-augmented data, and used for hyperparameter grid-search. Models performance is measured according to their weighted F1-scores on the test set.

The model is trained using Adam optimiser and cross entropy loss, using the weight argument to account for the class imbalance. The output dimension is 3,

which corresponds to the number of pT stages. To prevent overfitting, we use
L2 regularisation in the optimiser (weight decay parameter) and early stopping
based on the pseudo-validation dataset. We tune the following hyperparameters
using a grid search: dropout ratio {0, 0.5}, learning rate {10e-4, 10e-5, 10e-
6}, weight decay {10e-4, 10e-5}, number of hidden layers {16,32,64}, and batch
size {32, 64}. The values that provide the best performance in terms of class-
weighted F1 score correspond to a dropout ratio of 0.5, a learning rate of 10e-5,
a weight decay of 10e-5, 32 hidden layers, and a batch size of 64. The baseline
model consists of a multilayer perceptron which takes as input the individual
nuclei features (size, shape and marker expression) without taking the network
topology into consideration. The latent representations of these node features
are concatenated before the final layer to form a tile representation before the
final prediction.

**Post-hoc explainability** We compute Integrated Gradients [21] using the
model interpretability library for PyTorch Captum [11] to obtain an importance
score of individual edges and nodes for the pT stage prediction of each instance
in the test set. We can then compare areas of predictive importance across the
different selected RoI regions. The GNN Explainer model (implemented using
PyTorch Geometric) is used to obtain feature importances across all tiles in the
test set.

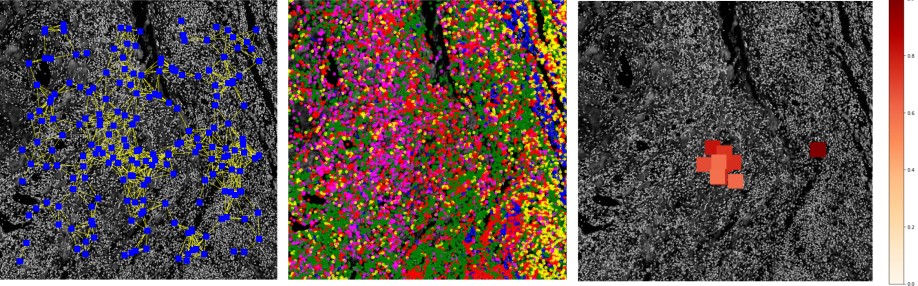

**Fig. 3.** An example of an invasive front RoI for a pT0 sample classified correctly. (Left)
Tile-graph of 200 256x256 tiles overlaid on DAPI. (Centre) Cell-graph corresponding to
the 2048x2048 RoI: blue - epithelial, green - T-helper, red: cytotoxic T-cell, magenta:
T-reg, yellow: B-cell. (Right) Top ten tiles classified as important using integrated
gradients for predicting tumour stage from immune interaction features.

**Results and discussion** As shown in Table 2, in the majority of the ex-
periments the invasive front was the region with the highest predictive power,
followed by the peritumoural stroma, known to have a high prognostic impact.
Moreover, all the graph-based models present an improvement over the baseline

model: this result suggests that the network topology plays an important role in tumour stage classification. Among the graph-based models, global max-pooling performed better than average pooling, as seen in [17]. Due to the limited number of samples with pT3, we were not able to correctly classify any of the pT3 RoIs. However, front pT3 RoIs were predicted to have pT2 stage, demonstrating that the model has learned to identify immune features related to an advanced cancer state. The proportion of interactions between FOXP3 positive and epithelial cells and the average expression of CD20 were found to be the top two features affecting tumour stage classification. Fig. 3 shows an example of tiles selected by IG as important in an invasive front RoI: it can be observed that the network considers a large cluster of regulatory T-cells as the most significant area for the prediction.

**Table 2.** Mean and standard deviation of RoI-level accuracies and class-weighted F1-scores measured on the test set and averaged over three distinct train-test splits.

| Model | GCN - Mean pool | | GCN - Add pool | | GCN - Max pool | | MLP baseline | |
|---|---|---|---|---|---|---|---|---|
| Region | Acc. (%) | F1(%) | Acc.(%) | F1(%) | Acc.(%) | F1(%) | Acc.(%) | F1(%) |
| All | $60.4_{\pm1.6}$ | $58.4_{\pm1.7}$ | $64.4_{\pm3.5}$ | $61.6_{\pm4.2}$ | $66.6_{\pm3.6}$ | $61.6_{\pm3.2}$ | $61.8_{\pm0.8}$ | $49.0_{\pm0.7}$ |
| Centre | $55.2_{\pm7.8}$ | $53.8_{\pm6.4}$ | $64.6_{\pm6.4}$ | $60.7_{\pm8.9}$ | $63.5_{\pm1.5}$ | $58.0_{\pm0.1}$ | $61.5_{\pm1.1}$ | $49.2_{\pm0.8}$ |
| Front | $68.8_{\pm5.1}$ | $63.6_{\pm8.5}$ | $66.7_{\pm5.9}$ | $60.4_{\pm9.9}$ | $\mathbf{72.9}_{\pm7.8}$ | $\mathbf{67.8}_{\pm9.0}$ | $62.4_{\pm0.3}$ | $48.9_{\pm0.7}$ |
| Mucosa | $49.8_{\pm11.4}$ | $47.5_{\pm12.6}$ | $56.3_{\pm5.9}$ | $50.2_{\pm8.1}$ | $63.1_{\pm9.6}$ | $60.8_{\pm9.9}$ | $62.1_{\pm2.7}$ | $48.4_{\pm1.9}$ |
| Stroma | $63.4_{\pm4.8}$ | $56.6_{\pm7.4}$ | $66.0_{\pm3.8}$ | $58.9_{\pm8.5}$ | $68.1_{\pm2.9}$ | $61.6_{\pm6.5}$ | $61.7_{\pm1.7}$ | $48.4_{\pm1.2}$ |

## 4   Conclusion

Our experiments demonstrate that the proposed two-layer GNN opens up new possibilities for interrogating multiplexed immuno-fluorescence images. As the model is capable of predicting tumour stage with a mean accuracy of well over 65%, we conclude that the model captures disease relevant information at a local level. The improvement with respect to the baseline model, which considers marker expressions and nuclei properties in isolation without accounting for their distribution and interaction, suggests that the network topology plays an important role in the tumour stage classification. Finally, in any given RoI, the post-hoc explainability method, together with the multi-tile strategy, allowed us to identify specific areas and features that contributed the most to the prediction. This will enable a follow up analysis to identify novel features that are of biological and clinical interest.

## Acknowledgement

The work presented in this paper has been partially supported by [***].

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
