# OpenReview forum: "Graph Based Neural Networks for Interrogating Multiplexed Tissue Samples"
_MICCAI.org/2021/Workshop/COMPAY — Reject_

### Official Review · Reviewer_eis8 · 2021-08-07
**The authors apply graph neural networks to multiplexed tissue data for predicting tumor stage**

**Rating:** 5
**Confidence:** 4

**Review:**

The authors propose the use of a 2 layer graph and a graph neural networks applied to hand crafted features extracted from mutiplexed data, the first layer of the a graph is constructed using cell graphs and the second layer of the graph is constructed using tile graphs where adjacency is defined by distance.

Though the idea is interesting the paper would benefit from clarity on whether the 2 layer graph (cell graph and tissue graph) are trained simultaneously or in stages, the section "Graph construction" the it implies that the graph is trained in two stages, if this is the case it is unclear what is being optimized in the first "message passing" phase. More detail in figure 1 would help with the overall understanding of the method.

Though providing explainability is meritable, Figure 3 should be expanded upon more, is this a cherry picked example? Are there examples of failure that are also of note?

The idea and application are interesting but more work needs to be done to make the contribution of the paper clearer.

---

### Official Review · Reviewer_HxKm · 2021-08-16
**Interesting method for a solved problem**

**Rating:** 5
**Confidence:** 3

**Review:**

In this study, the authors use a graph neural network (GNN) for image classification of multiplexed microscopy images. GNNs are a rather new technology that is only beginning to enter the computational pathology research community. From a technical standpoint, these networks are really interesting because they capture spatial interaction between cells, which convolutional neural networks do not analyze explicitly.  As far as I can tell, the methods are solid and innovative, although I cannot reproduce it because source codes are not provided. My main criticism is the target label which they train the GNN on: T-stage. T-stage can be defined by a pathologist in most cases just by looking at the glass slide without a microscope. In fact, this is so easy and fast to determine from H&E slides that it would make absolutely no sense to train a GNN on any non-routine image data (such as multiplexed data) to determine T-stage. It would make much more sense to try to predict N-stage or M-stage although these will be much harder to predict. So in summary, this is an interesting and innovative technology which should be applied to a clinically useful problem.

---

### Decision · Program_Chairs · 2021-08-25

Reject